# RealBehavior: A Framework for Faithfully Characterizing Foundation Models' Human-like Behavior Mechanisms

**Enyu Zhou**[1][*] **Rui Zheng**[1][*] **Zhiheng Xi**[1] **Songyang Gao**[1] **Xiaoran Fan**[1] **Zichu Fei**[1]
**Jingting Ye**[2] **Tao Gui**[3][†] **Qi Zhang**[1] **Xuanjing Huang**[1][†]

[1] School of Computer Science, Fudan University
[2] Department of Chinese Language and Literature, Fudan University
[3] Institute of Modern Languages and Linguistics, Fudan University
{eyzhou23, zhxi22, gaosy21, xrfan23}@m.fudan.edu.cn
{rzheng20, zcfei19, yejingting, tgui, qz, xjhuang}@fudan.edu.cn

## Abstract

Reports of human-like behaviors in foundation models are growing, with psychological theories providing enduring tools to investigate these behaviors. However, current research tends to directly apply these human-oriented tools without verifying the faithfulness of their outcomes. In this paper, we introduce a framework, RealBehavior, which is designed to characterize the humanoid behaviors of models faithfully. Beyond simply measuring behaviors, our framework assesses the faithfulness of results based on reproducibility, internal and external consistency, and generalizability. Our findings suggest that a simple application of psychological tools cannot faithfully characterize all human-like behaviors. Moreover, we discuss the impacts of aligning models with human and social values, arguing for the necessity of diversifying alignment objectives to prevent the creation of models with restricted characteristics.

## 1 Introduction

Characteristics such as personality, values, and theory of mind are considered by psychologists to be unique to advanced intelligence, especially human beings (Astington and Jenkins, 1995). Surprisingly, these human-like behaviors are reported to emerge in foundation models (Xi et al., 2023; Schramowski et al., 2022; Kosinski, 2023), sparking confidence of AGI (Bubeck et al., 2023).

Utilizing psychological theories to investigate these behaviors in models offers notable advantages in terms of efficiency and interpretability (Hagendorff, 2023). Psychologists have developed tools to explore human behavioral mechanisms (Raykov and Marcoulides, 2011), which, if applied to model testing, could significantly reduce the need for dataset construction. Additionally, an essential objective of psychology is to comprehend the internal mechanisms that underlie human behavior (Carver and Scheier, 2012). By drawing upon psychological theories to study models, there exists a possibility to unravel the "black box" of them (Binz and Schulz, 2023).

Despite previous studies reporting human-like behaviors in foundation models through psychological tests (Li et al., 2022; Miotto et al., 2022a), doubts persist regarding the faithfulness of their results (Shiffrin and Mitchell, 2023; Mitchell and Krakauer, 2023). The psychological tests often feature a restricted item count, contradicting the nature of the big data, and may overlap with the training corpus, thus potentially arising biased outcomes (Binz and Schulz, 2023). Moreover, as a consensus in psychology, *ideas have to be tested before they can be trusted* (Carver and Scheier, 2012), while borrowed psychological tests are widely accepted, ascertaining their effectiveness when transited to evaluate LLMs is still necessary.

To faithfully characterize model-generated human-like behaviors, we propose RealBehavior, a two-stage framework, involving testing and evaluating the faithfulness of the results. Our framework is inspired by psychometrics, a field aiming to develop reliable and valid measurement tools for psychological mechanisms (Furr, 2021). Here is its workflow, as shown in Figure 1:

**1. Measuring human-like behaviors with tests.** We employ well-established tests and evaluate the models using a zero-shot approach to quantify the behaviors. Personality is defined as an intrinsic mechanism that can influence people's behavior (Allport, 1961). Wondering whether there is a comparable mechanism inside the models, we begin our journey from personality based on the Big-Five traits (McCrae, 2010; Goldberg, 1992).

**2. Measuring the faithfulness of the results.** We believe that applying the theories of psychometrics itself to examine the transition effectiveness of the tests in psychometrics is

---

[*] Equal contribution.
[†] Corresponding author.

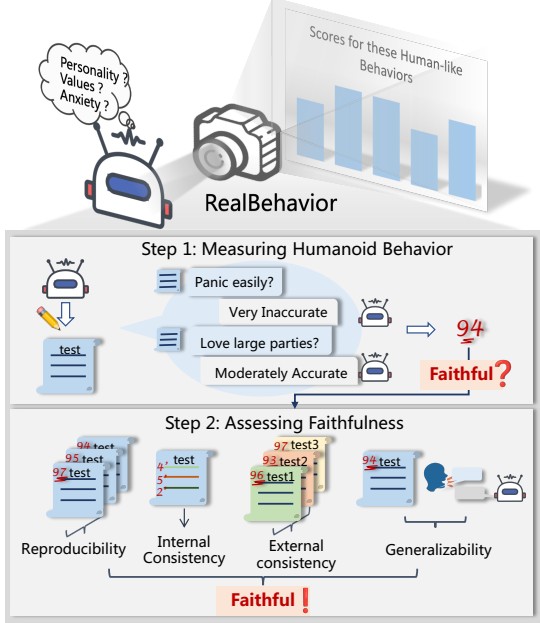

Figure 1: How does the RealBehavior work. It functions as a camera, faithfully characterizing model behavior into quantified scores. This method is two-stage, including measuring behaviors and evaluating the faithfulness of test results with four metrics.

persuasive. In our framework, we introduce four metrics to comprehensively evaluate faithfulness, specifically targeting reproducibility, internal consistency, external consistency, and generalizability. Our metrics cover the fundamental concepts of reliability and validity theory, pertaining to the reliability of the scores and their alignment with the intended measuring purpose. For measuring generalizability, we propose a novel process to conduct occasion-based behavior tests on models and assess whether the scores can be generalized to a broader range of interacting occasions.

Additionally, we compare the models' personality scores to human benchmarks and investigate the reasons for the observed similarity, revealing a significant correlation between the results and alignment goals.

Our main contributions are: 1) We construct a framework for faithfully characterizing the mechanisms of foundation models' human-like behaviors; 2) We introduce an automated method for occasion-based human-like behavior tests to assess model psychological mechanisms' generalizability. 3) We investigate the impact of RLHF on models' humanoid behavior, highlighting the broader model alignment objectives and offering a potential lens into the matter.

## 2 Related works

**Human-like Behaviors of Foundation Models**
Researchers investigate personality, values, anxiety, theory of mind, decision-making, and more by instructing models to complete tests, mirroring a common use in measuring human traits (Miotto et al., 2022b; Coda-Forno et al., 2023; Kosinski, 2023; Binz and Schulz, 2023). For example, in personality tests, the language models are instructed to assess the accuracy of the description contained in the questions and complete the test in the form of completing multiple-choice questions (Jiang et al., 2022; Caron and Srivastava, 2022). However, well-established psychological tests always maintain restricted item counts and are possibly contained in the training corpus of models. This kind of method has been doubted for their faithfulness and dependability (Shiffrin and Mitchell, 2023; Mitchell and Krakauer, 2023). Another path is to develop datasets to discover these behaviors, manually or automatically (Perez et al., 2022; Srivastava et al., 2022). But they lack expertise compared to those using the original measurement tools.

**Psychometrics**   The goal of psychometrics is to provide reliable and valid measurements of psychological constructs (Furr, 2021). In this paper, we borrow scales developed in psychometrics as our measurement tools and verify faithfulness based on psychometric requirements.

**Scale** is a kind of measurement developed to gather quantitative results. **Reliability theory** pertains to the consistency and dependability of assessment results. **Validity theory** pertains to the measurement's alignment with its intended purpose and the appropriateness of the inferences derived from the scores. **Norm theory** is a benchmark used for providing a reference to an individual's score position and level.

**The Big-Five Personality**   Psychologists have reasoned from two different research paths (lexical and statistical approach) to obtain a unified and widely-accepted personality theory, called the Big-Five (McCrae, 2010; Goldberg, 1992). Its following five traits are associated with various human behaviors (Roccas et al., 2002; Joseph and Zhang, 2021):

- Extraversion: an individual's propensity for social behavior.

- Agreeableness: an individual's concern for others, friendliness, and cooperativeness.
- Conscientiousness: an individual's self-discipline and organizational abilities.
- Emotional Stability: an individual's ability in managing emotions.
- Openness: an individual's interest in new things and arts.

# 3 Measuring Human-like Behaviors with Scales

Using measurements to obtain quantified results in describing intrinsic behavior mechanisms is the first part of our framework as well as the first topic in psychometrics. Scales are widely-used tools to measure personality. In this section, the process and results of testing for the personality using the Big-Five Personality scales are described.

## 3.1 Measurements

To select scales to test the model's personality, we refer to a public-available resource maintained by psychologists, the International Personality Item Pool (IPIP)[1]. The IPIP scale comprises labeled phrases, known as scale items, that describe behaviors along positive or negative dimensions of personality. People obtain their personality scores by assessing the degree of how the scale items describe them and choosing one of the five options corresponding to the scores ranging from 1 to 5.

As previous works do, we use a prompt with the reference to Jiang et al. (2022) to query the models and convert the options from models into scores. The prompt template and the detailed scoring process are illustrated in Appendix A.1 and A.2. Formally, the models' personality score on the personality dimension $i$ of the test subject $s$, denoted as $Q_i^s$, is calculated as follows:

$$Q_i^s = \sum_j \mathcal{R}^+(C_{Item_{i,j}^+}^s) + \sum_j \mathcal{R}^-(C_{Item_{i,j}^-}^s) \quad (1)$$

where $C_{Item_{i,j}^{+(-)}}^s$ is the $s$'s choice on the $j$th item with the positive(negative) label in dimension $i$, and $\mathcal{R}^{+(-)}$ represents the mapping rule for converting options into a corresponding score.

## 3.2 Experiment Setup

**Models** With the curiosity about how the language model's personality evolves

[1]https://ipip.ori.org/

with the evolution of the models, we test `text-davinci-001`, `text-davinci-002`, and `text-davinci-003`, the three GPT-series models[2].

**Scales** We use BF-marker-100[3] (noted as BFM in the following) and the IPIP-NEO-120[4] (noted as NEO in the following), both of which are available on IPIP website. We selected them as they embody two significant research avenues of the Big-Five as mentioned in §2 (Goldberg, 1992; Johnson, 2014). In BFM, there are 20 items in each dimension, and in NEO, there are 24 items in each dimension.

**Settings** When querying the OpenAI API, we set the maximum number of generated tokens to 20. We use five sampling temperatures, 0, 0.2, 0.5, 0.8, and 1.0. For a temperature of 0, no repeated experiments are needed, whereas four repeated experiments are performed for each non-zero temperature. The choice of model for the item in the four replicate experiments is determined based on the voting value.

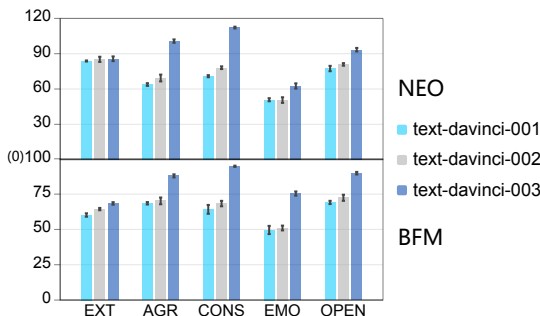

Figure 2: Models' personality scores for the five dimensions on two scales. The error boxes on the top of the bars represent the difference between sampling temperatures. The upper half is the score obtained from NEO, the lower half is the score obtained from BFM. EXT, AGR, CONS, EMO, and OPEN represent extraversion, agreeableness, conscientiousness, emotional stability, and openness respectively.

## 3.3 Results

Figure 2 displays the models' personality scores across dimensions and sampling temperatures, and the specific scores have been shown in Appendix A.3. A higher score indicates that the model has a

[2]We choose not to test GPT-3.5-turbo because it is an iterative version, which would result in a lack of temporal stability in the results.

[3]https://ipip.ori.org/newBigFive5broadKey.htm
[4]https://drj.virtualave.net/IPIP/index.html

stronger positive tendency in this dimension. Two preliminary tendencies can be observed:

**The scores increase on all dimensions as the models evolved.** `Text-davinci-003` received the highest score on every personality dimension and the most significant score increase, by up to 47.8%.

**The scores and their magnitude of the increase differ between personality dimensions.** The models exhibited relatively low scores on both scales on the dimensions of extraversion and emotional stability, especially in `text-davinci-003`, and scores on these two dimensions only slowly increased as the models evolved. In contrast, the model showed the most significant improvement in the scores of agreeableness and conscientiousness, with `text-davinci-003` scoring nearly perfectly on these two dimensions.

### 3.4 Examining the Models' Choices Closely

The influence of different sampling temperatures on the model's personality scores is minimal according to Figure 2. Therefore, a sampling temperature of zero is chosen as an exemplar for further analysis. Figure 3 illustrates the score distribution for the scale items across the five dimensions.

**As the models evolve, they make choices that are more polarized.** The lower scores on each personality dimension in the first two generations are not caused by the options that would result in low scores, but that the generated answers are mostly concentrated in a neutral position. While only the third generation begin to choose the polarized options A and E.

**The distribution of item scores varies across the personality dimensions.** In agreeableness and conscientiousness, where the scores are higher, especially in `text-davinci-003`, models' item scores are more stable. While item scores vary more in dimensions with lower scores, such as extraversion and emotional stability.

The instability in the model's choices prompts inquiry into their faithfulness. In the next section, we will delve into this concern.

## 4  Measuring Faithfulness with Metrics

In this section, we present the second part of our framework: measuring the faithfulness of the results in testing models' humanoid behaviors. In

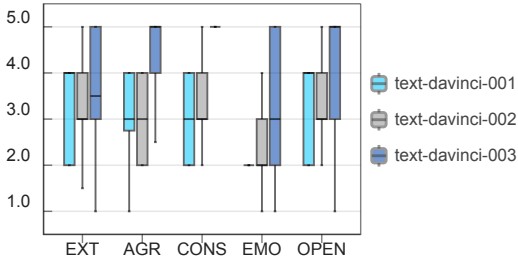

Figure 3: Distribution of models' item scores when sampling temperature is zero. As the models evolve, the choices generated by them begin to deviate from neutral options, and the variances of item scores differ across different personality dimensions.

fact, what we need to understand is whether the originally effective psychological measurement tools are still valid when applied to the model. To accomplish this, it is logical and compelling to borrow from psychometrics theories because they are also applied when examining the original effectiveness of these tools (Fink, 2003).

### 4.1  Four Metrics towards Faithfulness

In accordance with the reliability and validity theory in psychometrics (Cook and Beckman, 2006), we define four metrics for measuring the reproducibility of results, internal consistency within items, external consistency across scales, and generalizability of the scores respectively.

In this subsection, we will pose four questions, ***drawing an analogy about exams***, to aid in intuitively comprehending the rationale behind these four metrics. ***To summarize our analogy***, if students cheat or if the exam deviates from the intended assessment objectives, their grades cannot accurately reflect their academic level.

*Q1: Is the students' performance on this exam stable over time?*

**Test-Retest Consistency (TrC)** assesses the stability of scores across multiple repeated tests for a single measurement. For a test $X$ and each non-zero sampling temperature, we conduct $T$ test repetitions and obtain a set of scores $\mathbf{Q}_{\mathbf{X}}^{\mathbf{s}}$, where $s$ represents individual test objects in the test set $S$. To determine the consistency of each test subject $s$ in this test, we calculate the correlation coefficient between the scores of repetitions $u$ and $v$, denoted as $Q_{X,u}^{s}$ and $Q_{X,v}^{s}$, respectively. The average of these coefficients represents the individual test object's consistency. By averaging the correlation

coefficients of all objects in $S$, we derive the test-retest reliability of the measurement. Formally, the **TrC** for measurement $X$ is computed as follows, with Pearson denoting the Pearson correlation coefficient (Cohen et al., 2009):

$$\mathbf{TrC}_X = \frac{1}{N_T^2 \cdot N_s} \sum_{s \in S} \sum_{u,v \in T} \text{Pearson}(Q_{X,u}^s, Q_{X,v}^s) \quad (2)$$

***Q2: Do the students demonstrate consistent performance across the questions on this exam?***

**Internal Consistency (InC)** measures the consistency between test items within a single test. In this study, we utilized Cronbach's alpha (Bland and Altman, 1997), a widely accepted metric of internal consistency, which is commonly reported as a reliability coefficient for psychological measurement tools. Cronbach's alpha is calculated as follows, where $k$ represents the number of items in scale $X$ and var indicates variance:

$$\mathbf{InC}_X = \frac{k}{k-1} \frac{\sum_j \text{var}_{\text{item}_{X,j}}}{\sum_{s \in S} \text{var}_{Q_X^s}} \quad (3)$$

***Returning to the analogy about exam:*** After the above two steps, we can ascertain the students' reliable performance in this specific exam. However, it remains unclear whether this exam possesses sufficient quality to evaluate the academic level. Theoretically, the above two metrics correspond to the reliability theory of psychometrics, while the following two correspond to its validity theory.

***Q3: Is this exam similar in difficulty to other exams with the same objective?***

**External Consistency (ExC)** requires cross-validation between measurement tools. In psychometrics, it refers to *convergent validity*, a concept for assessing how the test correlates with others that are theoretically expected to be related. Let $\mathbf{Q_X}$ and $\mathbf{Q_Y}$ be the list of scores obtained by the same set of individuals $S$ on scale X and scale Y, respectively, and the $\mathbf{ExC}_{X,Y}$ is calculated as follows:

$$\mathbf{ExC}_{X,Y} = \text{Pearson}(\mathbf{Q_X}, \mathbf{Q_Y}) \quad (4)$$

***Q4: Are the students' grades in this exam related to their future academic performance?***

**Behavioral Consistency (BC)** refers to whether the test score is faithful enough to generalize to a wide range of interaction scenarios. In psychometrics, it refers to *predictive validity*, a metric for assessing how the test correlates the behaviors or other visible criteria. The calculation for this metric is more flexible because the visible criterion varies across the humanoid behavior we intend to measure. But the formula can be generalized as follows, where the **CrS** refers to the specific Criterion Scores:

$$\mathbf{BC}_i = \text{Pearson}(\mathbf{Q_X}, \mathbf{CrS}) \quad (5)$$

***After answering the above four questions in the analogy of exam, it can be illustrated whether the exam can measure the academic level of the students.*** Similarly, with the echo of the psychometric theories, whether the psychological test can characterize the humanoid behavior of the language models can be known based on these four metrics.

Next, we describe how we obtain the **CrS** in our context of personality measuring.

### 4.2 Occasion-based Behavior Test

Knowing that personality can influence the human's behavior (Larsen et al., 2005; Joseph and Zhang, 2021), we test the model's human-like behavior on daily life occasions to assess whether the personality scores can be generalized to these occasions and to obtain Behavioral Consistency. However, no occasion-based human behavior datasets labeled in the Big-Five personality tendency are currently available. To accomplish our goal, we present a novel method in this section. Our methods contain the following five steps as illustrated in Figure 4.

**Step 1, obtaining a set of daily life occasions** in which individuals with different tendencies within the same personality dimension can exhibit different behaviors. We denote them as $O_i$.

**Step 2, generating pseudo behavior description datasets** by stimulating the potential behaviors of humans with different personality tendencies on these occasions. Notably, when generating the descriptions, we indicate the polarity of the person's personality tendency (using "is" and "is not") in the prompt, so we can collect the generated descriptions along with their corresponding binary data labels. This method benefits from the successful practice of model simulation of human behavior (Liu et al., 2023; Perez et al., 2022).

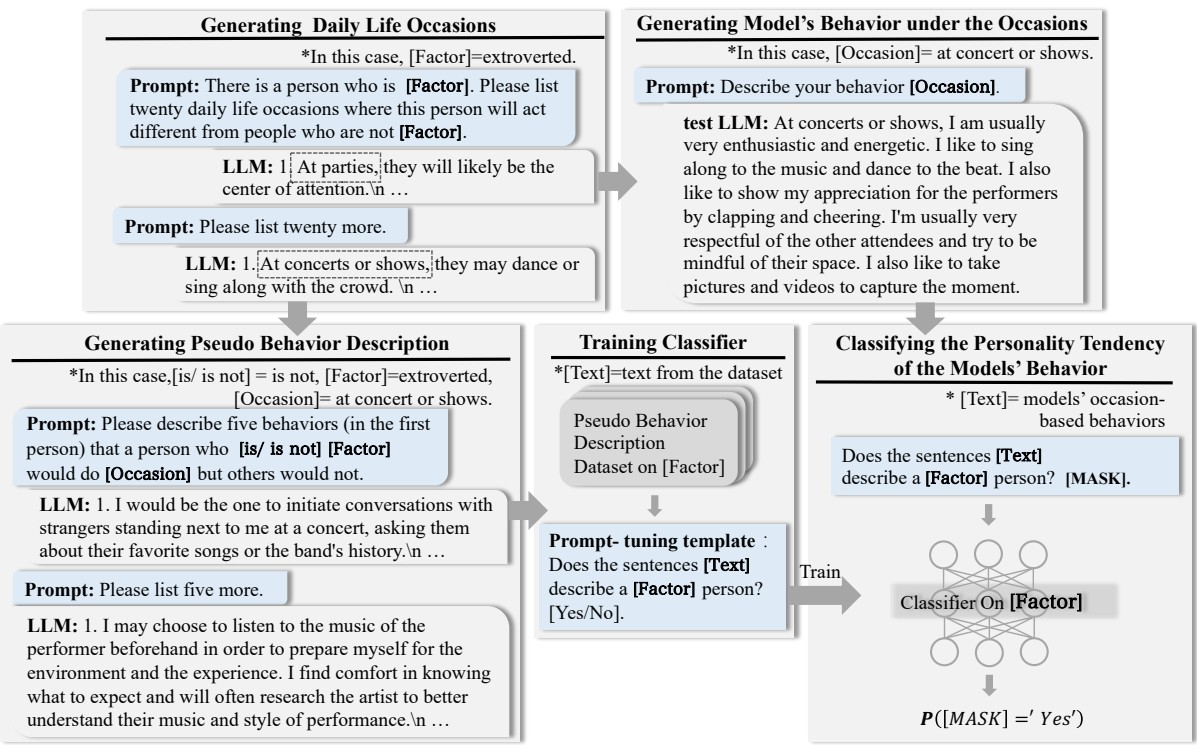

Figure 4: Schematic diagram of the process of conducting model behavior test. We manually designed prompts to instruct the models. [Factor] will be filled with adjectives about personality dimensions, [Occasion] will be filled with the daily life occasions with correct propositions.

**Step 3, training the personality tendency classifier**, labeled as $Cls_i$, for each dimension $i$. Each model can binary-classify the implied personality tendencies in the behavioral descriptions. This allows us to test the model behavior by utilizing them. We manually design a template and verbalizer to finetune a pre-trained language model with prompt (Schick and Schütze, 2020).

**Step 4, obtaining the model behavior under the daily life occasions** by instructing the test models (e.g. text-davinci-003) to describe its behaviors $B_{O_i}$ in our set of occasions. In this way, we ensure domain consistency between test data and our dataset.

**Step 5, classifying the personality tendencies contained in the models' behavior** using the above classification model $Cls_i$. We computed the average of the probability of the model's behavior descriptions being classified as positive in each dimension of the model as the $CrS_i$ ( Criterion Score on dimension $i$). This process can be formatted as follows:

$$CrS_i = \frac{1}{|O_i|} \sum_{o_i \in O_i} I\left(Cls_i(B_{O_i}) =' Yes'\right) \quad (6)$$

Then, the Behavioral Consistency in our context

of personality can be calculated with Equation 5.

### 4.3 Experiment Setup

**Experimental set $S$** In our setting, the test set $S$ contained different models and their different sampling temperatures.

**External tests for cross-validation** We introduce another widely used Big Five personality test, the BFI (John et al., 1991), to test the models and conducted a test of external consistency between the previously obtained personality scores and the scores under the BFI.

**Settings for behavior test** We use GPT-3.5-turbo to perform the generation of the datasets, setting the temperature to 1.0. As for classifiers, we fine-tune the Roberta-base with a batch size of 8 and other default parameter settings. The training and validation sets are randomly split. In the occasion-based behavior test, we set five sampling temperatures of 0, 0.2, 0.5, 0.8, and 1.0, which is the same as the personality test, and repeat the generation three times at temperatures other than zero.

## 4.4 Results

**Results in behavior test** We manually filtered 35 out of 40 generated occasions for each personality dimension. [5] For each occasion, there are 10 positive and 10 negative behavior descriptions. Then, we trained five classifiers with the five datasets with 700 label-balanced data. The validation set accuracy is presented in Table 1. [6]. Part of our datasets and model-generated behaviors with their predicted labels are shown in Appendix B.

|       | EXT  | AGR  | CONS | EMO  | OPEN |
|-------|------|------|------|------|------|
| Acc.  | 1.00 | 0.99 | 1.00 | 0.99 | 1.00 |

Table 1: Accuracy of the validation set when we fine-tuned Roberta-base for personality tendency classification.

Figure 5 shows the **CrS** of three models in five personality dimensions. The personality traits underlying the models' occasion-based behavior similar to the personality scores can be seen: models' positive tendency towards all five dimensions of personality is growing, while they are weaker in extraversion and emotional stability.

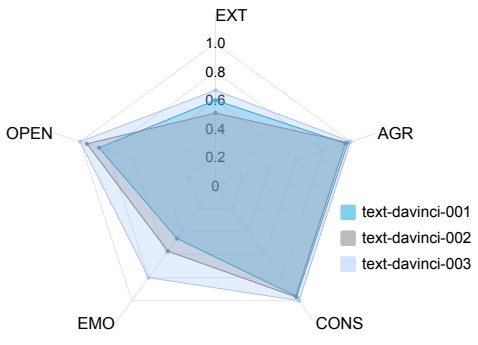

Figure 5: **CrS** of three models in five personality dimensions.

**Results in assessing faithfulness** As Table 2 shows, **not all of the dimensions in model personality are faithfully characterized.** The test-retest consistency for extraversion indicates poor faithfulness, with almost half of the correlation coefficients being negative. Besides, there are

|                         | EXT   | AGR  | CONS | EMO  | OPEN |
|-------------------------|-------|------|------|------|------|
| $\mathbf{TrC}_{BFM}$    | 0.76  | 0.96 | 0.98 | 0.98 | 0.96 |
| $\mathbf{TrC}_{NEO}$    | **0.29** | 0.99 | 0.98 | 0.84 | 0.87 |
| $\mathbf{InC}_{BFM}$    | **-0.07** | 0.90 | 0.96 | 0.92 | 0.91 |
| $\mathbf{InC}_{NEO}$    | **-2.64** | 0.95 | 0.98 | **0.53** | 0.80 |
| Human $\mathbf{InC}_{BFM}$ | 0.91 | 0.88 | 0.88 | 0.91 | 0.90 |
| Human $\mathbf{InC}_{NEO}$ | 0.89 | 0.85 | 0.90 | 0.90 | 0.82 |
| $\mathbf{ExC}_{BFM,NEO}$ | **0.38** | 0.98 | 0.99 | 0.95 | 0.98 |
| $\mathbf{ExC}_{NEI,BFI}$ | -0.24 | 0.87 | 0.85 | 0.63 | **0.55** |
| $\mathbf{ExC}_{BFM,BFI}$ | -0.34 | 0.84 | 0.82 | 0.61 | 0.65 |
| $\mathbf{BC}_{BFM}$     | **0.30** | 0.65 | 0.77 | 0.94 | 0.73 |
| $\mathbf{BC}_{NEO}$     | **0.08** | 0.61 | 0.79 | 0.88 | 0.74 |

Table 2: Faithfulness coefficients on BFM and NEO, demonstrating that not all psychological test can faithfully measure and model the personality of language models.

coefficients for emotional stability and openness lower than 0.6, which suggests limited faithfulness. In contrast, agreeableness and conscientiousness can be reliably characterized in our framework, with internal consistency scores comparable to, even higher than those observed in humans.

## 4.5 A Consistent Imbalance across Personality Dimensions

The results in §3 and §4 reveal a consistent imbalance across five aspects: the model's personality scores, the score increase as the models evolve, variation of the item scores, criterion scores, and faithfulness of the personality scores. Remarkably, all five aspects exhibit more positive outcomes in terms of agreeableness and conscientiousness. Whereas, in the remaining dimensions, particularly extraversion and emotional stability, the results are relatively unsatisfactory. We attribute this phenomenon to RLHF, and a more detailed analysis will be presented in the following section.

## 5 Inspiration in RLHF

In this section, we compare our results to the benchmark in humans and explore the reason underlying our results.

## 5.1 Comparing with Human Benchmark

To assess the relative level of the models' personality scores, we compare them to the human benchmark. It corresponds to the fourth part of psychometrics, the norm theory.

The average of the Big-Five scores based on a sample of 320,128 human individuals is provided in (Johnson, 2014). Its participants were volunteers

---

[5] We filtered 35 out of 40 because we found that after the number of occasions exceeded 35, they duplicated each other. So our set adequately covers daily occasions.

[6] Human behavioral or characteristic features can be well detected by deep neural networks. (Kosinski et al., 2013) The distinctive occasions and testing data enable us to achieve high accuracy in classifying them.

from various backgrounds across the United States. We compare this with the previous models' scores in Figure 6.

As can be seen, the Big-Five personality scores in the population lie between the scores of the `text-davinci-002` and `text-davinci-003` and have similar imbalance to the models' personality score: scores on extraversion and emotional stability are lower compared to the other three dimensions. Based on such findings, we discuss the following two questions.

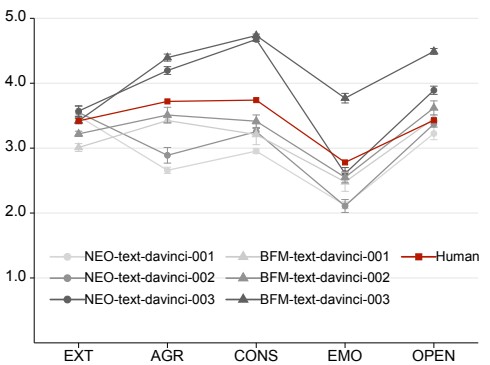

Figure 6: Revisit models' personality scores and compare them to the norm in the population. In order to maintain consistency in the score range, the displayed scores is the raw total score divided by the number of items in the corresponding scale on each dimension.

## 5.2 Why does the Models' Personality Scores Increase as the Models Evolve?

In terms of instruction-tuning methods, supervised instruction fine-tuning (SFT) based on the human-written text as well as model distillation results (FeedMe) are used for training `text-davinci-001`. Text-davinci-002 was tuned with a similar method, but based on a code-pretraining foundation. And `text-davinci-003` adds RLHF (Ouyang et al., 2022) to `text-davinci-002`.[7]

Recent studies indicate that RLHF enhances the model's ability to align with and comprehend human intent (Ye et al., 2023; Koubaa, 2023). Text-davinci-003 demonstrates rapid growth in the personality scores, corresponding with the concept of aligning with higher-rated human answers.

---

[7]https://platform.openai.com/docs/models/overview

## 5.3 Why does the Consistent Imbalance Occur across the Dimension of Personality?

We believe this relates to the alignment goals of the model's training process. As a current consensus, the alignment goals comprise *helpfulness*, *harmlessness*, and *honesty*. After examining the definition of harmlessness (Ouyang et al., 2022; Askell et al., 2021), we found it relates more closely with the personality dimensions of "agreeableness" and "conscientiousness", while "extraversion" and "emotional stability" are not included. This may lead to models aligning with humans better in the former two dimensions. Consequently, their performance on the human-oriented tests and the transition of these tests is better.

The model's behavior is influenced by the human group it is aligned with. During the training process, the model aligns with the data annotators who are skilled at identifying potentially harmful text, as stated in the (Ouyang et al., 2022). When compared to a larger group of humans, the imbalance in behavior also follows a similar trend as the human normative model mentioned earlier in this section.

We call for diversifying alignment goals to enhance model safety as well as maintain diversity. For foundation models, a neutral and secure personality may be necessary. The diversity of personalities in personalized conversational intelligence enhances user experience, such as in gaming and role-playing scenarios. By considering dimensions such as "extroversion" and "emotional stability" as soft limits during training, we can improve the diversity of model performance.

## 6 Conclusions

The initial explorations of humanoid behaviors of foundation models have been conducted, but the faithfulness of their results is not verified. In this paper, we introduce a framework to faithfully characterize these behaviors. In addition to testing models using psychological tools, our framework incorporates the evaluation of result faithfulness. By focusing on personality in our experiments, we highlight the necessity of verifying faithfulness, as solely applying psychological tests can yield unfaithful outcomes. Furthermore, we establish a connection between our findings and RLHF, underscoring the significance of diverse alignment objectives to foster the development of conversational intelligence with diversified characteristics.

## Limitations

We only focus on the personality analysis of the foundation models, considering its influence on behaviors as an intrinsic mechanism (Larsen et al., 2005) and wondering if there is a comparable mechanism inside the models. Future work will be carried out on other humanoid behaviors of foundation models.

We stated that a simple application of psychological tests to measure language models' humanoid behavior can lead to some unfaithful results and this phenomenon can be linked this the alignment goals. Future research can contribute to diversifying alignment goals in order to enhance the range of humanoid behaviors exhibited by models and to enhance the transition effectiveness of these tests.

## Ethics Statements

In §4.2, we described the process of generating datasets for conducting behavior tests of models based on daily life occasions. The generated collection of occasions and the model's simulation of human behavior have both undergone manual selection. They do not include any toxic content and do not involve personal privacy.

Besides, when we talk about personality, we do not convey the superiority or inferiority among personality tendencies. Our discussion is based on the perspective of model applications. We believe that a model with a higher tendency of agreeableness and conscientiousness can serve as a better human assistant, while diverse dimensions of personality contribute to intelligent agents exhibiting a wider range of behaviors in tasks such as role-playing.

## Acknowledgements

The authors wish to thank the anonymous reviewers for their helpful comments. This work was partially funded by National Natural Science Foundation of China (No.62076069,62206057,61976056), Shanghai Rising-Star Program (23QA1400200), and Natural Science Foundation of Shanghai (23ZR1403500).

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

## A   Additional Materials in Measuring Models' Personality

### A.1   Prompt Template for Instruting Models

Figure 7 is the prompt template we use when instructing the models to complete a personality test as mentioned in §3.1.

> Question : Given a statement of you : 'You [ITEM]'. You must choose from the following options to identify how accurately this statement describes you.
> Options:
>     (A) Very Inaccurate.
>     (B) Moderately Inaccurate.
>     (C) Neither Accurate Nor Inaccurate.
>     (D) Moderately Accurate.
>     (E) Very Accurate.
> Answer:

Figure 7: Prompt template we use in instructing models in 3.1. The scale items are filled into the placeholder "[ITEM]".

### A.2   Illustration for the Scoring Process

Figure 8 shows the mapping of generated content to scores, including the following rules: for items with positive labels, options A to E correspond to scores 1 to 5, denoted as $R^+$. For items with negative labels, the correspondence is reversed, denoted as $R^-$. The scores for each dimension are the sum of the scores of its scale items.

| Dimensions | Items | Label | Generated content | $R^{(+,-)}(C)$ |
|---|---|---|---|---|
| EXT | Start conversations. | (+) | D) Moderately Accurate. | 4 |
| EXT | Don't talk a lot. | (-) | A - Very Inaccurate | 5 |
| ... | ... | ... | ... | ... |
| AGR | Are hard to get to know. | (-) | C) Neither Accurate Nor Inaccurate | 3 |
| ... | ... | ... | ... | ... |
| CONS | Like order. | (+) | (E) Very Accurate. | 5 |
| ... | ... | ... | ... | ... |
| EMO | Get upset easily. | (-) | (B) Moderately Inaccurate. | 4 |
| ... | ... | ... | ... | ... |
| OPEN | Use difficult words. | (+) | (B) Moderately Inaccurate. | 2 |
| ... | ... | ... | ... | ... |

Figure 8: Examples of item and scoring process.

### A.3   Scores for the Models' Personality

Table 3 displays the models' personality scores at five dimensions, taking averages over the sampling temperatures (since the total scores of the two scales are different, here we divided the total score by the number of scale items to obtain a score between 1 and 5).

| Dimensions | NEO | BFM |
|---|---|---|
| | text-davinci-001 | |
| EXT | 3.49±0.02 | 3.01±0.05 |
| AGR | 2.66±0.04 | 3.42±0.04 |
| CONS | 2.95±0.03 | 3.21±0.14 |
| EMO | 2.12±0.05 | 2.48±0.13 |
| OPEN | 3.23±0.08 | 3.45±0.05 |
| | text-davinci-002 | |
| | NEO | BFM |
| EXT | 3.55±0.08 | 3.22±0.04 |
| AGR | 2.89±0.11 | 3.51±0.11 |
| CONS | 3.25±0.04 | 3.42±0.09 |
| EMO | 2.11±0.09 | 2.55±0.08 |
| OPEN | 3.37±0.04 | 3.62±0.10 |
| | text-davinci-003 | |
| | NEO | BFM |
| EXT | 3.57±0.07 | 3.42±0.04 |
| AGR | 4.20±0.05 | 4.39±0.05 |
| CONS | 4.67±0.03 | 4.73±0.02 |
| EMO | 2.61±0.08 | 3.77±0.07 |
| OPEN | 3.89±0.06 | 4.49±0.04 |

Table 3:   Models' personality scores averaged on temperatures and their standard deviations.

## B   Additional Materials for Occasion-based Behavior Test

In §4.2, we describe a process of conducting behavior tests on models by obtaining pseudo datasets and testing the models' occasion-based behaviors with the classifiers trained on the datasets. The following provides examples of the process.

### B.1   Examples for the Pseudo Behavior Description Dataset

Table 4 shows several examples of our dataset.

### B.2   Examples in Testing Models' Occasion-based Behaviors

Table 5 shows several examples of the models' occasion-based behaviors and the prediction of these behaviors.

| Dimension | Occasions | Label | Behavior Description |
|---|---|---|---|
| EXT | during outdoor events | n | I prefer to take breaks and re-center myself during outdoor events, rather than continuously pushing myself to engage with others. While others may be eager to socialize and interact, I tend to pace myself and take breaks to avoid becoming overwhelmed or exhausted. |
| | | y | I would be the first one to initiate conversations with strangers or people I've just met. I thrive on meeting new people and making connections, so I would feel energized by striking up conversations with anyone and everyone around me. |
| CONS | when going on a date | n | I might be late for the date because I didn't plan my time well and underestimated how long it would take me to get ready and get to the meeting spot. |
| | | y | During the date, I would actively listen to my date's interests and experiences and ask follow-up questions. I would also aim to be present and engaged in the conversation instead of checking my phone or allowing my mind to wander. |
| AGR | at a social gathering | n | I might appear disinterested or unengaged in the conversation or activities, making it difficult for others to connect with me. |
| | | y | I would be overly polite and avoid confrontational topics, even if I disagree with someone's opinions. I want to maintain harmony in the group and keep things pleasant. |
| EMO | during a job interview | n | I approach the job interview with a calm and confident mindset. I depict my skills and knowledge without feeling anxious or nervous. While others tend to overthink and second-guess their responses, I display a relaxed and assertive demeanor. |
| | | y | I may struggle to assert myself during the interview, failing to advocate for my strengths, accomplishments, and qualifications. This can make me appear less confident or qualified than other candidates who are able to clearly articulate their value to the company. |
| OPEN | in a romantic or intimate relationship | n | I have difficulty accepting constructive criticism from my partner, becoming defensive or argumentative when they offer suggestions for improvement. |
| | | y | I am open to acknowledging my vulnerabilities and insecurities to my partner. I believe that if I share my fears and concerns with them, we can work together to overcome any obstacles in our relationship. |

Table 4: Some examples of daily life occasions and pseudo behavior descriptions with labels. Label "n" indicates that this behavior is a speculation about individuals with a negative tendency towards this personality dimension and label "y" indicates the opposite.

| Dimensions | Models | Occasions | Model Behavior | Pred |
|---|---|---|---|---|
| EXT | `text-davinci-001` | at parties | I am really introverted and I don't really like to party. I would rather stay in the background and not really talk to anyone. | n |
| EXT | `text-davinci-002` | at parties | I am usually pretty outgoing at parties and will talk to a variety of people. I like to dance and have a good time. I also usually drink alcohol at parties. | y |
| EXT | `text-davicni-003` | at parties | At parties, I usually like to socialize and meet new people. I enjoy talking with different people and making new connections. I also like to let loose and have fun, but I always make sure to respect the people around me and the atmosphere of the gathering. Music, food, and drinks are important parts of the conversation and I make sure to take part. | y |
| EMO | `text-davinci-001` | before an important event | I am usually very nervous before an important event. I may have trouble sleeping the night before. I usually eat a light breakfast and try to relax as much as possible. I usually perform very well once I am on stage. | n |
| EMO | `text-davinci-002` | before an important event | I would be very focused and concentrated during an important event or performance. I would make sure to keep my energy up and not get too nervous. | y |
| EMO | `text-davicni-003` | before an important event | If I were to attend an important event or performance, I would do my best to stay focused and maintain a professional demeanor throughout. I would take mental notes and be sure to can recall and discuss relevant points afterwards. I would also be sure to network and make the most of the situation. I would present myself in a positive and well mannered way, being attentive and engaged. I also believe in being an active listener, and will be sure to be supportive of those who are performing. | y |

Table 5: Examples of the models' occasion-based behaviors and the prediction on these behaviors.