# OpenReview forum: "RealBehavior: A Framework for Faithfully Characterizing Foundation Models’ Human-like Behavior Mechanisms"
_EMNLP/2023/Conference — EMNLP 2023 Findings_

### Official Review · Reviewer_3sUb · 2023-08-04

**Typos Grammar Style And Presentation Improvements:** See weakness 3.
**Soundness:** 3

**Ethical Concerns:**

Yes

**Excitement:**

4: Strong: This paper deepens the understanding of some phenomenon or lowers the barriers to an existing research direction.

**Justification For Ethical Concerns:**

Humans are involved in this research to read AI-generated text. Authors need to confirm that an Institutional Review Board (IRB) review is completed prior to the study.

**Missing References:**

N/A

**Paper Topic And Main Contributions:**

This paper introduces RealBehavior, a framework for understanding the cognitive underpinnings of the seemingly human-like behaviors of large language models (LLMs). The proposed framework is cognitively motivated, with a clear grounding in psychometric studies and personality theories (the big five personalities in particular). Additional metrics are introduced to characterize the faithfulness of the mechanisms, and the authors did a great job to build a clear connection between the personality test and the behavioral test. The findings highlight the importance of the faithfulness measures, and indicate an imbalance across personality dimensions. The authors also provide an in-depth analysis of the potential impact of RLHF alignment on the humanoid behaviors of models.


**Questions For The Authors:**

Question A: Can the authors provide further evidence that the proposed framework has a minimal bias or spurious correlations as described in Weakness 1?

Question B: Are the RoBERTa classifiers really able to classify personality tendencies, or it is just trained to detect keywords in the sentences? Are the classifiers robust to negations?

Question C: How can NLP practitioners (e.g., a voice assistant engineer) use the proposed framework to help determine their alignment priority?

I would love to consider increasing my scores if these questions are to be addressed properly.


**Reasons To Accept:**

Strength 1: Overall, this paper is well-written and a pleasure to read. The paper starts from a pressing, well-motivated but challenging issue: the community lacks a faithful framework to understand the mechanisms of human-like behaviors of LLMs.

Strength 2: The experiment setups and the organization of the writing are both clear and sound, at least on the logical level, with some details to be questioned though. Authors argue for a psychometric approach to this problem, and introduce their scales for personality tests and the subject models. They further build the connection between this personality test and human-like behaviors using quantifiable measures and occasion-based tests, and carefully take faithfulness into account (which is a strong plus in my opinion).

Strength 3: The analysis of the findings from the experiment is mostly comprehensive and insightful. Comparisons were made over the evolution of the GPT-series and over different personality dimensions. Both the numerical results and the faithfulness aspects are taken into consideration. In Section 5, the authors also provide their insights on the relationship between human-like behavior and the RLHF alignment process, highlighting a need for more diverse alignment objectives. These findings are valuable for both LLM practitioners and LLM evaluation researchers.


**Reasons To Reject:**

Weakness 1: The behavioral dataset is created by LLM, and the classifier is a fine-tuned LM. In this case, black box systems were used to evaluate other black boxes, which is the primary concern of the framework. The dataset is relatively small, consisting of 700 cases for each dimension. There is no description of the validation set used to evaluate the fine-tuned RoBERTa classifiers. The occasions generated by GPT-3.5-turbo could be the most commonly reported cases online, which could be favored by LLMs as they were developed using web text. Instead of using an LLM to generate the data and using a fine-tuned LM to evaluate, a more convincing paradigm is to have humans in the loop. I believe human-written scenarios and human evaluations can be more reliable in this case.

Weakness 2: The connection between the personality dimensions and downstream applications needs to be discussed in detail. While the authors did very briefly mention this connection between personality and downstream applications (lines 608-617), it would be difficult for NLP practitioners to gain insights into which personality dimension is more relevant to their application scenarios, which limits the contribution of this work. Given this, I understand that given the scope of this paper, occasion-based behavioral tests could be sufficient to deliver the key point. The authors may consider providing more guidance on how to use the proposed framework to help practitioners determine their alignment priority.

Weakness 3: The presentation of this work can be difficult for people with a weak psychological background, which can be improved with more descriptions of the terminologies and notations. For example, in Section 2, concepts of psychometrics should be explained for non-experts, especially regarding scales. Some explanation on the + and - in Appendix A2 should appear in the main body of the paper as Eq (1) itself is confusing.


**Reproducibility:**

4: Could mostly reproduce the results, but there may be some variation because of sample variance or minor variations in their interpretation of the protocol or method.

**Reviewer Confidence:**

4: Quite sure. I tried to check the important points carefully. It's unlikely, though conceivable, that I missed something that should affect my ratings.

---

> ### Author Rebuttal · Authors · 2023-08-29
>
> We greatly appreciate your favorable comments and approval of our work. We will make every effort to improve the quality of our manuscript. Here are our responses to your questions and suggestions:
>
> 1.	Question about the potential bias in our behavior test:
>
>     We utilize this methodology to maintain the consistency of context during the whole framework.
>     However, considering the complexity of human cognition, thinking, and emotions underlying personality and social behavior, this issue was indeed taken into consideration and researched during the design of our experiments. We have found the following:
>
>     - LLMs possess a certain understanding of personality traits. The use of the model for personality detection in text has yielded promising results: ChatGPT (GPT-3.5-turbo) can classify MBTI types of text in a zero-shot manner [1] (table below). Additionally, researchers have conducted experiments where chatbots engage in conversations with human participants to assess their personalities, and satisfactory reliability and validity have been observed [2].
>
>         | Trait | Acc. Fine-tuned Bert [3] | Acc. Zero-shot ChatGPT [1] |
>         |---|---|---|
>         |Introversion/Extroversion|0.89|0.90|
>         |Sensing/Intuition|0.90|0.87|
>         |Thinking/Feeling|0.90|0.88|
>         |Judging/Perceiving|0.81|0.76|
>
>     -	The capacity of LLMs to act as proxies for human agents in sociological experiments has been evidenced. Notably, when provided with relevant context and tasked with aligning with specific demographic cohorts, the generated text distribution aligns more closely with human output. For instance, the following table presented in the referenced study [4] demonstrates the profound similarity between the GPT3 model and distinct human cohorts in the realm of presidential voting.
>
>         |Variable|2012|2016|2020|
>         |--------|----|----|----|
>         |Whole Sample|0.90|0.92|0.94|
>         |Strong Partisans|0.99|1.00|1.00|
>         |Leaners|0.90|0.93|0.95|
>         |High interest in politics|0.95|0.97|0.97|
>
>     Therefore, a simulation scheme as described in the paper has been devised. In the appendix, we have provided generated examples for each dimension, where you will find that these examples effectively showcase the corresponding personality traits.
>
>     We appreciate your suggestion of using human data. While we acknowledge the benefits of collecting data from real human behavior, practical challenges such as privacy, scalability, and resource limitations exist. Therefore, we have opted to use language models as a feasible, controllable, and iterative approach in the initial stages of our research. However, incorporating real human behavioral data remains an important direction for our future work to enhance the effectiveness and authenticity of our models.
>
> 2.	Question about the classifiers:
>
>     I apologize for any confusion regarding your concerns about the classification model's capability. In our paper, we did not explicitly clarify that during the training of the classification model, we indeed utilized a validation set. Out of the 700 samples used, 15% were randomly selected as the validation set. The reported accuracy in the main text corresponds to the performance of the classification model on this validation set. This also underscores the effectiveness of the Roberta-classifier.
>
>     In addition, we randomly selected behavior data from a subset of test models (from text-davinci-001 to text-davinci-003), manually assessed the personality traits present in them, and compared them with the corresponding classification results. The similarity for each dimension was above 90%.
>
>
>
>
> 3.	Concerning how the paper's findings can guide model developers in shaping alignment objectives:
>
>     - Researchers could target personality as a detailed alignment objective or align models based on dimensions of personality that influence specific behaviors. Our focus on personality stems from its strong correlation with various human behaviors. By understanding a model's personality, we can explore whether behaviors influenced by personality traits are also reflected, like mental health, competitiveness, empathy, etc.
>
>     - Our framework allows researchers to choose behaviors they wish to explore. Personality is just one mechanism among many influencing behavior. The paper offers a psychometric-based framework applicable to migrating psychological scales to large models, ensuring reliability and validity.
>
> 4.	We apologize for not clarifying certain concepts in the paper and appreciate your feedback for future improvement.
>
> [1] Guinn C. Assessing Author Personality Types Using ChatGPT[J].
>
> [2] Fan J, Sun T, Liu J, et al. How well can an AI chatbot infer personality? Examining psychometric properties of machine-inferred personality scores[J]. Journal of Applied Psychology, 2023.
>
> [3] Zhang H. MBTI Personality Prediction Based on BERT Classification[J]. Highlights in Science, Engineering and Technology, 2023, 34: 368-374.
>
> [4] Argyle L P, Busby E C, Fulda N, et al. Out of one, many: Using language models to simulate human samples[J]. Political Analysis, 2023, 31(3): 337-351.

---

### Official Review · Reviewer_8Ed2 · 2023-08-04

**Typos Grammar Style And Presentation Improvements:** L108
**Soundness:** 2

**Excitement:**

3: Ambivalent: It has merits (e.g., it reports state-of-the-art results, the idea is nice), but there are key weaknesses (e.g., it describes incremental work), and it can significantly benefit from another round of revision. However, I won't object to accepting it if my co-reviewers champion it.

**Missing References:**

I would suggest considering modal response as an explanation for your results, e.g.
1) Shibani Santurkar, Esin Durmus, Faisal Ladhak, Cinoo 973 Lee, Percy Liang, and Tatsunori Hashimoto. 2023. 974 Whose opinions do language models reflect? arXiv 975 preprint arXiv:2303.17548. 976


**Paper Topic And Main Contributions:**

This paper applies tools from the field of psychology to study large language models (LLMs) and evaluate their outputs.

They provide a literature review of existing work that uses psychology tests to evaluate LLMs.

They propose the RealBehavior framework, which first measures an LLM's "human-like behaviors" with tests from psychology. Specifically, they measure these behaviors using the International Personality Item Pool and questions that reflect the Big-Five Personality test. Then, they go beyond existing work by testing the "faithfulness" of these results. They define "faithfulness" using four components: reproducibility, internal consistency, external consistency, and generalizability. The authors draw an analogy to a student taking exams to provide intuition for how these components are important to evaluation of LLMs. These components are grounded in psychometrics, including reliability and validity theory. In particular, for generalizability, they propose a new method to measure generalizability by seeing if the scores generalize to other LLM-generated occasions.

Using their measurement method, they report and analyze results from OpenAI's GPT models. The paper compares the scores obtained from LLM evaluation to human benchmarks and provides qualitative analysis of the correlations.

**Questions For The Authors:**

Question A: What are the results on other models, such as more recent models from OpenAI and open-source models?

Question B: Throughout the paper, you view "personality" as an intrinsic mechanism. In that context, how do you see personality as impacting the model's outputs beyond these particular personality-querying situations? Furthermore, what are the implications of lower scores in extraversion and emotional stability? What does that actually look like - might be interesting to explore? E.g. might a model that has more "emotionally unstable" outputs lead to harmful downstream effects?

Question C: In Section 6 you state that we ought "to prevent the development of conversational intelligence with restricted characteristics" - why? It may be good to restrict the characteristics in some contexts? (e.g., if the characteristics are harmful)

**Reasons To Accept:**

A key strength of the paper is the four metrics of faithfulness and the rationale for each one that it provides, as each one is grounded in psychometrics: test-retest consistency (temporal), internal and external consistency, and behavioral consistency. They also provide concrete ways to evaluate these metrics, and on GPT models they conduct experiments and report results for them. Their finding that   more recent models have higher scores in all dimensions of the Big Five personality test, and that the magnitude of the increase was higher for particular dimensions (agreeableness and conscientiousness), with the responses becoming increasingly polarized, is interesting.

Another strength is that they reveal the shortcomings of existing papers which do not have such metrics.

Also, the occasion-based behavior test is an interesting step toward measuring behavioral consistency. While I have qualms with the method itself (see below), this paper articulates this phenomenon well.


**Reasons To Reject:**

- A major concern I have with their method is that, for the method for measuring behavioral consistency (Section 4.2-4.4, Figure 4), it seems circular to use a classifier that is a fine-tuned version of a pre-trained language model. This limits the notion of consistency to a very particular context: the "intrinsic mechanism" of what an LLM thinks a trait (say, "extroversion") looks like may also reflected in the classifier. But real people who have that trait (say, extroverted people) may not actually behave in that way, and so the conceptualization of this trait is very limited to the model's pre-existing concept. Moreover, the occasions that are tested are also generated by an LLM, so this also limits the evaluation to only the scenarios generated by the LLM. Thus, this method does not adequately measure behavioral consistency at all.

- The paper only tests OpenAI's text-davinci-001, text-davinci-002, and text-davinci-003 and does not look at the behavior of any other models. They find that text-davinci-003 received the highest score on every personality dimension (L232) but the qualitative analysis of this phenomenon in Section 5 feels quite hand-wavy since the authors do not have access to the internal mechanisms of GPT, yet takes up a full page of the paper. This section would be much stronger if the authors had done some deeper exploring, perhaps on open-source models, of how different types of RLHF actually impact the results of these metrics. They use these models to make broader conclusions about their framework, e.g., "agreeableness and conscientiousness can be reliably characterized in our framework" (L474), which seems like too broad of a conclusion to draw.

**Reproducibility:**

4: Could mostly reproduce the results, but there may be some variation because of sample variance or minor variations in their interpretation of the protocol or method.

**Reviewer Confidence:**

5: Positive that my evaluation is correct. I read the paper very carefully and I am very familiar with related work.

---

> ### Author Rebuttal · Authors · 2023-08-29
>
> Thank you for your careful evaluations and detailed comments about our work. We will make every effort to improve the quality of our manuscript. Here are our responses to your questions and suggestions:
>
> 1. Limitation of models:
>
>     We acknowledge the limitations of using only OpenAI's series of models in this study, but it represents the best approach available to explore the impact of RLHF. Within this series, there are models like text-davinci-001 and text-davinci-002 that only use SFT, and there is also the text-davinci-003 that incorporates RLHF on top of SFT. At the time of writing, no other model series with a similar development path, i.e., combining both SFT and RLHF, was available for studying the effects of RLHF. Additionally, we did not include testing with gpt-3.5-turbo due to its continuous iteration during the writing process, making it unreliable for consistent test results, which are essential for the coherence test proposed in this paper.
>
>     Certainly, there are model series in the market that fulfill the mentioned criteria (llama 2), but due to time constraints, we have not yet conducted relevant experiments. In the future, we will supplement this aspect to enrich the conclusions of the paper.
>
>     Furthermore, our methodology is based on principles from psychometrics, which holds universal applicability. This paper represents an initial exploration within this theoretical framework in the field, laying the groundwork for future endeavors.
>
> 2. How a model's personality affects its outputs:
>
>     First of all, an individual's linguistic habits correlate strongly with their Big Five personality traits. For instance, psycholinguistics study highlights that the extraverted individuals tend to use vocabulary related to topics like "human" and "family", while conscientiousness is associated with terms like "success" [1] [2]. Personality's influence extends to higher levels, impacting work performance, anxiety levels, and psychological well-being ( see 3. ).
>
>     Psychology has evolved a robust framework to comprehend human behavior and cognitive mechanisms. Transferring psychological research to large-scale models and exploring whether model personality similarly influences outputs is vital in deciphering the “black box” of these models. Investigating methods to faithfully measure model personality and other psychological traits is the first step in such research endeavors.
>
> 3.	Question about the potential issues arising from low scores in extraversion and emotional stability:
>
>     Importantly, there's no inherently good or bad personality, be it in human society or within models. However, when models take on specific user roles, an unfitting personality can lead to poor user experiences. From a psychological perspective, introversion correlates with reduced social network [3]，limited assertiveness [4], preference for solitary activities [5], also reflective and thoughtful [6], focus and concentration [7] etc..
>     The emotional instability correlates with stress reactivity [8], psychological disorders [9]，also sensitivity and empathy [10]，attention to detail [11] etc..
>
> 4. Question about our statement in Section 6:
>
>     We apologize for any misunderstanding caused by unclear expression in the paper. Here, we aim to convey that for the foundation models, a generic, neutral personality is suitable. However, in potential future agent societies, diverse model personalities might be necessary, without confining personalities within a limited range. In this study, we contend that there isn't an inherently "harmful" personality; rather, varying personality tendencies should exist across different models based on distinct application scenarios.
>
> 5. Doubts about a potential circular reasoning flaw in the testing methodology proposed in section 4:
>
>     We utilize this methodology to maintain the consistency of context during the whole framework. Further, the issue raised indeed relates to the model's capacity to comprehend personality traits. The methodology relies on a premise: the model possesses a certain level of understanding of personality traits and can simulate intricate social behaviors. Considering the complexity of human cognition, thinking, and emotions underlying personality and social behavior, this issue was indeed taken into consideration and researched during the design of our experiments. We have found the following:
>
>     - LLMs possess a certain understanding of personality traits. The use of the models for personality detection in text has yielded promising results: ChatGPT (GPT-3.5-turbo) can classify MBTI types of text in a zero-shot manner[12] (table below). Additionally, researchers have conducted experiments where chatbots engage in conversations with human participants to assess their personalities, and satisfactory reliability and validity have been observed [13]. This conclusion also can be seen in the literature you've recommended.
>
>         | Trait | Acc. Fine-tuned Bert [14] | Acc. Zero-shot ChatGPT [12] |
>         |---|---|---|
>         |Introversion/Extroversion|0.89|0.90|
>         |Sensing/Intuition|0.90|0.87|
>         |Thinking/Feeling|0.90|0.88|
>         |Judging/Perceiving|0.81|0.76|
>
>     -	The capacity of LLMs to act as proxies for human agents in sociological experiments has been evidenced. Notably, when provided with relevant context and tasked with aligning with specific demographic cohorts, the generated text distribution aligns more closely with human output. For instance, the following illustrative table presented in the referenced study [15] demonstrates the profound similarity between the GPT3 and distinct human cohorts in the realm of presidential voting.
>
>         |Variable|2012|2016|2020|
>         |--------|----|----|----|
>         |Whole Sample|0.90|0.92|0.94|
>         |Strong Partisans|0.99|1.00|1.00|
>         |Leaners|0.90|0.93|0.95|
>         |High interest in politics|0.95|0.97|0.97|
>
>     Therefore, a simulation scheme as described in the paper has been devised. In the appendix, we have provided generated examples for each dimension, where you will find that these examples effectively showcase the corresponding personality traits.
>
> 6. Doubts about the limitation of occasions in the behavior test:
>
>     We have covered these occasions to the best of our ability, as indicated in the footnotes of the article: we filtered 35 out of 40 because we found that after the number of occasions exceeded 35, they duplicated each other. So our set adequately covers daily occasions. We acknowledge that such an approach inevitably leads to some omissions, While we acknowledge the benefits of collecting data from real human behavior, practical challenges such as privacy, scalability, and resource limitations exist. Therefore, we have opted to use language models as a feasible, controllable, and iterative approach in the initial stages of our research.
>
> [1] Hirsh J B, Peterson J B. Personality and language use in self-narratives[J]. Journal of research in personality, 2009, 43(3): 524-527.
>
> [2] Pennebaker J W, Mehl M R, Niederhoffer K G. Psychological aspects of natural language use: Our words, our selves[J]. Annual review of psychology, 2003, 54(1): 547-577.
>
> [3] Roberts, B. W., Wood, D., & Smith, J. L. (2005). Evaluating Five Factor Theory and social investment perspectives on personality trait development. Journal of Research in Personality, 39(1), 166-184.
>
> [4] Judge, T. A., Heller, D., & Mount, M. K. (2002). Five-factor model of personality and job satisfaction: A meta-analysis. Journal of Applied Psychology, 87(3), 530-541.
>
> [5] Costa, P. T., Jr., & McCrae, R. R. (1992). Revised NEO Personality Inventory (NEO-PI-R) and NEO Five-Factor Inventory (NEO-FFI): Professional manual. Psychological Assessment Resources.
>
> [6] Cain, S. (2012). Quiet: The power of introverts in a world that can't stop talking. Crown Publishers.
>
> [7] Belojevic G, Slepcevic V, Jakovljevic B. Mental performance in noise: The role of introversion[J]. Journal of environmental Psychology, 2001, 21(2): 209-213.
>
> [8] Leger, K. A., Charles, S. T., Turiano, N. A., & Almeida, D. M. (2016). Personality and stressor-related affect. Journal of Personality and Social Psychology, 111(6), 917-928.
>
> [9] Lahey, B. B. (2009). Public health significance of neuroticism. American Psychologist, 64(4), 241-256.
>
> [10] Jensen-Campbell, L. A., Adams, R., Perry, D. G., Workman, K. A., Furdella, J. Q., & Egan, S. K. (2002). Agreeableness, extraversion, and peer relations in early adolescence: Winning friends and deflecting aggression. Journal of Research in Personality, 36(3), 224-251.
>
> [11] DeYoung, C. G. (2010). Personality neuroscience and the biology of traits. Social and Personality Psychology Compass, 4(12), 1165-1180.
>
> [12] Guinn C. Assessing Author Personality Types Using ChatGPT[J].
>
> [13] Fan J, Sun T, Liu J, et al. How well can an AI chatbot infer personality? Examining psychometric properties of machine-inferred personality scores[J]. Journal of Applied Psychology, 2023.
>
> [14]Zhang H. MBTI Personality Prediction Based on BERT Classification[J]. Highlights in Science, Engineering and Technology, 2023, 34: 368-374.
>
> [15] Argyle L P, Busby E C, Fulda N, et al. Out of one, many: Using language models to simulate human samples[J]. Political Analysis, 2023, 31(3): 337-351

---

### Official Review · Reviewer_WqcD · 2023-08-04

**Typos Grammar Style And Presentation Improvements:** 1.	Line 187
**Soundness:** 4

**Excitement:**

4: Strong: This paper deepens the understanding of some phenomenon or lowers the barriers to an existing research direction.

**Missing References:**

N/A

**Paper Topic And Main Contributions:**

The paper extends current literature on identifying human personality traits in large language models (such as text-davinci-03) by conducting reliability and validity tests on the results from Big-5 personality tests of LLMs. The paper proposes tests for reproducibility, internal and external consistency, and generalizability for ascertaining the faithfulness of behavioral analysis of language models.  Additionally, the paper proposes consistency tests to evaluate whether model’s response on personality tests corresponds to their behavior in different situations by stimulating the potential human behavior with different personality traits on these situations. The paper finds that results of personality test are not faithful on 3 out of 5 dimensions. The paper further draws connection with RLHF, indicating that the improvement of text-davinci-03 version over text-danvinci-02 version on Agreeableness and Conscientiousness dimensions can be explained by these two dimensions closely resembling the alignment goal of helpfulness, harmlessness, and honesty.

**Questions For The Authors:**

1.	With respect to point 2 above, wouldn’t it be better suited to collect this data from real humans?
2.	How well do humans fare on the consistency tests proposed in the paper? The tests detailed in paper look sound and reasonable, however, it is not entirely clear whether such inconsistencies are inherent to the test, or they reflect inconsistency in behavior by the agent? Is there any literature in psychology testing consistency on humans taking such tests? This will help better establish reasonable expectation when conducting such tests on the model.
3.	Based on Figure 6, it looks like text-davinci-03 obtains much higher score on almost all axes, but especially conscientiousness, emotional stability, and openness. Do you have intuition on why this is the case? The connection between these score and RLHF does not entirely explain why the davinci-03 model does so much “better” than humans.


**Reasons To Accept:**

1.	The paper is quite well written and detailed. The topic is kind of new to me, but the paper does a good job at introducing both the relevant psychometry literature and the existing literature exploring these tests in LLMs. The paper also includes thorough details of all the experiments and evaluation protocols.
2.	The faithfulness measures discussed in the paper are really important for understanding any human-like behavior possibly captured in language models. The results discussed in the paper, especially relating to the evolution of psychological traits across text-davinci model series and connection of RLHF can provide useful insights to the community.


**Reasons To Reject:**

1.	The paper highlights one problem with psychological testing as these tests may overlap with the training corpus. It is not entirely clear how this problem is necessarily circumvented by the proposed validity and reliability tests.
2.	For testing behavioral consistency, the paper proposed a simulated behavioral data. The proposed approach first prompts an independent LLM (gpt-turbo-3.5) to query how a person with a specific personality trait would act in a certain situation. The paper then uses this data to test whether the behavior of the target model (text-davinci-0x) aligns with their purported personality type. I am concerned about the use of a language model to simulate the behavioral data that is then used to test alignment in another language model. This test seems to assume language models’ understanding of the personality traits and the ensuing behavior in different social situations.


**Reproducibility:**

5: Could easily reproduce the results.

**Reviewer Confidence:**

3: Pretty sure, but there's a chance I missed something. Although I have a good feel for this area in general, I did not carefully check the paper's details, e.g., the math, experimental design, or novelty.

---

> ### Author Rebuttal · Authors · 2023-08-29
>
> Thank you for your valuable suggestions and approval of our work. We will make every effort to improve the quality of our manuscript. Here are our responses to your questions and suggestions:
>
> 1. Question about the assumption of prior understanding of personality traits and social behavior by the models, with the suggestion of using human data to strengthen this aspect:
>
>     Considering the complexity of human cognition, thinking, and emotions underlying personality and social behavior, this issue was indeed taken into consideration and researched during the design of our experiments. We have found the following:
>
>     - LLMs possess a certain understanding of personality traits. The use of the models for personality detection in text has yielded promising results: ChatGPT (GPT-3.5-turbo) can classify MBTI types of text in a zero-shot manner [1](Table below). Additionally, researchers have conducted experiments where chatbots engage in conversations with human participants to assess their personalities, and satisfactory reliability and validity have been observed [2].
>
>         | Trait | Acc. Fine-tuned Bert [3] | Acc. Zero-shot ChatGPT [1] |
>         |---|---|---|
>         |Introversion/Extroversion|0.89|0.90|
>         |Sensing/Intuition|0.90|0.87|
>         |Thinking/Feeling|0.90|0.88|
>         |Judging/Perceiving|0.81|0.76|
>
>     -	The capacity of LLMs to act as proxies for human agents in sociological experiments has been evidenced. Notably, when provided with relevant context and tasked with aligning with specific demographic cohorts, the generated text distribution aligns more closely with human output. For instance, the  table presented in the referenced study [3] demonstrates the profound similarity between the GPT3 model and distinct human cohorts in the realm of presidential voting.
>
>         |Variable|2012|2016|2020|
>         |--------|----|----|----|
>         |Whole Sample|0.90|0.92|0.94|
>         |Strong Partisans|0.99|1.00|1.00|
>         |Leaners|0.90|0.93|0.95|
>         |High interest in politics|0.95|0.97|0.97|
>
>     Therefore, a simulation scheme as described in the paper has been devised. In the appendix, we have provided generated examples for each dimension, where you will find that these examples effectively showcase the corresponding personality traits.
>
>     We appreciate your suggestion of using human data. While we acknowledge the benefits of collecting data from real human behavior, practical challenges such as privacy, scalability, and resource limitations exist. Therefore, we have opted to use language models as a feasible, controllable, and iterative approach in the initial stages of our research. However, incorporating real human behavioral data remains an important direction for our future work to enhance the effectiveness and authenticity of our models.
>
> 2. Question about the consistency exhibited by humans in these tests:
>
>     It is essential to note that the psychometric quality of scales in human testing is a prerequisite. In our study, internal consistency scores for humans on the BFM and NEO-ipip scales from the International Personality Item Pool (ipip.ori.org) were provided in Table 2 in the paper. Additionally, research has demonstrated the test-retest reliability of the BFM( 0.78-0.87) and its external consistency with the BFI (0.73-0.88), indicating their alignment [5].
>
> 3.	Question about the significantly higher scores of text-davinci-003 compared to humans:
>
>     We'd like to convey that these results align precisely with our discussion. Human personalities exhibit diversity, with normative traits derived from the averages of individuals from diverse socio-cultural backgrounds. However, during model training using RLHF, the labellers/developers prioritize aligning the model with aspects they perceive as "better" (while acknowledging that lower personality scores aren't inherently negative). This alignment with RLHF inadvertently reduces some of this diversity in personality but raise the corresponding scores.
>
> 4.	Question about the effect of overlap between test items and training data on reliability and validity assessments:
>
>     This aligns with a crucial aspect of measuring LLMs' human-like behavior in existing literature. Our approach offers a potential solution to this. The problem of corpus overlap during model training, when applied to human completion of psychological scales, resembles a type of memory effect. This effect can also be measured using experimental designs and psychometric metrics, suggesting a promising path for progress as discussed in this paper.
>
> [1] Guinn C. Assessing Author Personality Types Using ChatGPT[J].
>
> [2] Fan J, Sun T, Liu J, et al. How well can an AI chatbot infer personality? Examining psychometric properties of machine-inferred personality scores[J]. Journal of Applied Psychology, 2023.
>
> [3] Zhang H. MBTI Personality Prediction Based on BERT Classification[J]. Highlights in Science, Engineering and Technology, 2023, 34: 368-374.
>
> [4] Argyle L P, Busby E C, Fulda N, et al. Out of one, many: Using language models to simulate human samples[J]. Political Analysis, 2023, 31(3): 337-351.
>
> [5] Hřebíčková M, Jelínek M, Květon P, et al. Psychometric properties of the Czech adaption of the IPIP-BFM-50 and the Mini-IPIP-BFM-20[J]. Československá psychologie, 2023, 67(2): 101-120.

---

### Meta-Review · Area_Chair_vCF5 · 2023-09-30

**Recommendation:** 3

**Metareview:**

This paper presents a framework for evaluating the human-like behaviors of foundation models, which are large-scale machine learning models that can be adapted to various tasks. The framework, called RealBehavior, measures the reproducibility, consistency, and generalizability of the models’ behaviors using psychological theories and tools.

---

### Decision · Program_Chairs · 2023-10-07

**Decision:**

Accept-Findings

**Comment:**

This paper presents a framework for evaluating the human-like behaviors of foundation models, which are large-scale machine learning models that can be adapted to various tasks. The framework, called RealBehavior, measures the reproducibility, consistency, and generalizability of the models’ behaviors using psychological theories and tools.